# Towards Agents That Know When They Don't Know: Uncertainty as a Control Signal for Structured Reasoning

## Abstract

Large language model (LLM) agents are increasingly deployed in structured biomedical data environments, yet they often produce fluent but overconfident outputs when reasoning over complex multi-table data. We introduce an uncertainty-aware agent for query-conditioned multi-table summarization that leverages two complementary signals: (i) retrieval uncertainty—entropy over multiple table-selection rollouts—and (ii) summary uncertainty—combining self-consistency and perplexity. Summary uncertainty is incorporated into reinforcement learning (RL) with Group Relative Policy Optimization (GRPO), while both retrieval and summary uncertainty guide inference-time filtering and support the construction of higher-quality synthetic datasets.

On multi-omics benchmarks, our approach improves factuality and calibration, just less than tripling correct and useful claims per summary (3.0→8.4 internal; 3.6→9.9 ulti-omics) and substantially improving downstream survival prediction (C-index 0.32→0.63). These results demonstrate that uncertainty can serve as a control signal—enabling agents to abstain, communicate confidence, and become more reliable tools for complex structured-data environments.

## 1 Introduction

Imagine a biomedical researcher querying a large multi-omics database to identify candidate biomarkers for survival outcomes Jin et al. (2024). A standard LLM-based agent may confidently produce a fluent statement such as "gene X is strongly associated with survival in patients"—even when the underlying tables contain contradictory or insufficient evidence. To the end user, this confident but unqualified claim is indistinguishable from a reliable finding Omar et al. (2025); Martell et al.. By contrast, an uncertainty-aware agent could detect the inconsistency, flag its own low confidence, or abstain altogetherZhao et al. (2025; 2024); Hu et al. (2024); Han et al. (2024). This ability to communicate not only what is said but also how certain it is transforms raw text generation into actionable, trustworthy scientific insight Bolton et al. (2024); Omar et al. (2025); Hakim et al. (2024).

Most modern scientific knowledge is encoded not in natural language but in high-dimensional tables such as genomic assays, proteomic screens, and electronic health records (Consortium, 2020; Bycroft et al., 2018; Kang et al., 2022; Probst & Reymond, 2020). These resources contain invaluable information that could accelerate biomedical discovery, yet they remain largely inaccessible to non-specialists. Extracting meaningful insights from such data, i.e. generating summaries, demands not only computational power but also the ability to translate complex numerical signals into coherent narratives—an area where LLMs are uniquely positioned to contribute (Li et al., 2025; Yu et al., 2025). The novelty of our work lies in using uncertainty-aware signals to both calibrate agents and filter summary outputs, enabling their use as synthetic data (Lee et al., 2025). This approach enhances the quality of training corpora, ultimately enabling more robust and reliable downstream decision-making.

Recent work has begun adapting LLMs for tabular summarization and reasoning. Query-focused methods such as QTSumm (Zhao et al., 2023) generate targeted textual insights from structured

inputs, while StructText (Kashyap et al., 2025) and eC-Tab2Text (Guanilo et al., 2025) introduce synthetic benchmarks across scientific and e-commerce domains. Evaluation frameworks such as FineSurE (Song et al., 2024) and multi-agent debate approaches (Estornell & Liu, 2024) reveal the challenges in measuring faithfulness and coverage in generated summaries, highlighting the limitations of current single-pass generation methods Sui et al. (2025).

An emerging paradigm involves designing table agents—LLM-driven systems that integrate structured querying, strategic planning, and external tool use Bendinelli et al. (2025); Mathur et al. (2024); Stoisser et al. (2025b). For example, Lu et al. (2025) outline design principles for real-world table agents capable of combining SQL execution with reasoning chains, while demonstrate multi-agent orchestration for multi-document reasoning tasks Sui et al. (2025). Beyond summarization, frameworks such as MAG-V (Sengupta et al., 2024) exemplify iterative generation and verification of synthetic data, illustrating a blueprint for refinement over one-shot output.

However, these promising approaches share a critical blind spot: uncertainty. LLMs are known to produce fluent yet unfaithful outputs (Xu et al., 2024), a problem exacerbated when summarizing high-dimensional data (Fang et al., 2024; Wu et al., 2025). We conceptualize uncertainty quantification (UQ) as a form of agent–environment interaction (Han et al., 2024), where the focus is not only on data quality but also on the agent's confidence and reliability in navigating complex tables. Recent efforts in UQ range from confidence–consistency scoring methods such as CoCoA (Vashurin et al., 2025) to head-based uncertainty prediction (RAUQ, UQLM (Vazhentsev et al., 2025; Bouchard et al., 2025)). Other works explore faithfulness-aware UQ in retrieval-augmented generation (Fadeeva et al., 2025) and structured tasks such as text-to-SQL (Somov & Tutubalina, 2025), underscoring the necessity of calibration for trustworthy table understanding.

In this paper, we propose an uncertainty-aware LLM agent for summarizing high-dimensional tabular data. Our agent generates candidate summaries from multi-omics datasets, quantifies its own uncertainty, and filters outputs with high uncertainty. We evaluate the approach on biomedical multi-omics tasks, where multiple valid summaries exist—highlighting the critical role of calibration beyond mere coverage.

Our contributions are threefold:

1. **Uncertainty as control:** We introduce the first LLM agent framework where uncertainty is not just monitored but directly used as a reward signal during training, and as an abstention/filtering signal at inference, moving beyond post-hoc diagnostics.

2. **Robustness in structured environments:** On biomedical multi-omics tasks, uncertainty-aware agents achieve higher factuality, calibration, and downstream utility, with methods applicable to any multi-table setting.

3. **Uncertainty as data-quality signal:** We show that filtering high-uncertainty samples improves tabular text dataset quality, providing a practical tool for curating reliable corpora.

## 2 BACKGROUND

### 2.1 INTERACTIVE AGENT FRAMEWORKS FOR STRUCTURED REASONING

Early table summarization methods primarily relied on rule-based or statistical approaches, producing template-based outputs and lacking explicit uncertainty modeling. Recent advances employ neural and LLM-based methods that shift from static, single-pass generation to interactive reasoning over structured environments. For example, LLM agents can now issue SQL queries or dataframe operations Stoisser et al. (2025a;c), dynamically retrieving evidence before forming summaries. Surveys of table agents Tian et al. (2025) highlight how symbolic querying and neural reasoning can be combined to support exploratory analysis and hypothesis generation. More recently, multi-agent frameworks such as MAG-V (generator–verifier) (Sengupta et al., 2024) and Multi2 (scalable multi-document reasoning)(Cao et al., 2025) demonstrate how dividing labor among specialized agents can improve reliability and scalability. These works suggest that interactive, tool-augmented agents are a promising direction for table understanding.

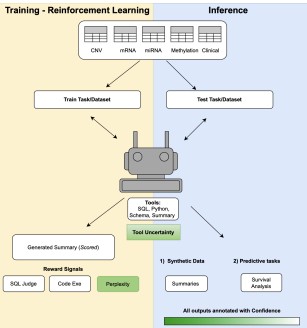

Figure 1: The uncertainty-aware agent framework. this diagram shows the two phases of our agent: (a) training with reinforcement learning, and (b) inference. in training, the agent's policy is refined using a reward signal informed by summary uncertainty (perplexity). during inference, multiple rollouts generate candidate summaries, which are then filtered based on a combined score of retrieval and summary uncertainty, leading to more reliable outputs.

## 2.2 UNCERTAINTY QUANTIFICATION IN LLMS

Despite progress in interactivity, most agents remain prone to overconfidence and unfaithful outputs. Traditional metrics such as BLEU or ROUGE fail to capture factual reliability in structured domains. This has led to the development of uncertainty quantification approaches and libraries such as CoCoA (Vashurin et al., 2025) and LM-Polygraph (Fadeeva et al., 2023), which use probabilistic confidence and/or semantic self-consistency to detect hallucinations. In structured tasks like text-to-SQL (Shorinwa et al., 2025), confidence estimation has been shown to prevent execution errors by flagging low-confidence predictions Maleki et al. (2025). Similarly, in retrieval-augmented generation, uncertainty-aware thresholds can trigger additional retrieval or abstention (Fadeeva et al., 2025; Soudani et al., 2025). However, most of these methods treat uncertainty as a post-hoc diagnostic Hao et al. (2025). They are not integrated into the agent's decision-making process during interaction with tables, limiting their effectiveness in dynamic environmentsHu et al. (2025).

## 2.3 TOWARD SELF-ASSESSMENT IN SCALABLE AGENTS

A growing body of work suggests that scalable and trustworthy agents must go beyond post-hoc uncertainty estimation toward learned self-assessment Han et al. (2024); Guan et al. (2024); Renze & Guven (2024). Active learning studies (Melo et al., 2024; Ye et al., 2025) show that focusing on uncertain cases improves efficiency, while debate-style multi-agent systems (Yin & Wang, 2025) demonstrate how structured disagreement enhances reliability. Recent explorations of self-reflection in LLMs indicate that agents can improve reasoning by monitoring their own confidence Liu et al. (2025a). Yet, existing work has not combined these insights into a framework where uncertainty directly controls both training optimization and inference-time behavior Cui et al. (2025); Zhang et al. (2025). Structured domains such as databases, where repeated querying and summarization are natural, provide fertile ground for such uncertainty-aware self-assessment. This paper builds on these insights by proposing a framework in which retrieval stability and output consistency are treated as first-class control signals, enabling LLM agents to produce more reliable and trustworthy multi-table summaries.

## 3 METHODS

We cast query-conditioned multi-table summarization as an episodic agent problem and make *uncertainty* a control signal: We (i) measure retrieval instability and output inconsistency, (ii) shape training rewards with those signals, and (iii) apply them during inference to filter summaries and enrich them with a quality signal.

## 3.1 PROBLEM FORMULATION

Let $\mathcal{D}$ be a structured database and $q$ a natural-language task. A policy $\pi_\theta$ interacts with $\mathcal{D}$ via tools and emits a summary $s$ that encapsulates the information in the database relevant to the given task:

$$(q, \mathcal{D}) \xrightarrow{\pi_\theta} s.$$

## 3.2 ENVIRONMENT AND EPISODE SETUP

Each episode takes place in an environment consisting of: (i) a structured database $\mathcal{D}$ containing tables, columns, and descriptions, and (ii) a task $q$. At timestep $t$, the state $x_t$ includes the task $q$, the schema snapshot of $\mathcal{D}$, and the history of previous actions and results. The agent selects actions $a_t \sim \pi_\theta(a_t \mid x_t)$, which the environment executes deterministically. Available actions are:

- **SQLExecutor(query)** – Executes a SQL query to retrieve or join rows across tables in $\mathcal{D}$.
- **Schema(table)** – Returns the structure, column names, and types of a specified table.
- **PythonTool(code)** – Runs Python code to process query results or perform computations when SQL is insufficient.
- **CommitSummary(summary)** – Terminates the episode and outputs a final summary $s$.

**Episode flow.** An episode thus consists of a query, a sequence of tool calls, and a terminating summary. Formally, invoking `CommitSummary` yields a trajectory

$$\tau = \big((x_0, a_0), (x_1, a_1), \ldots, (x_T, a_T)\big)$$

and a final output $s$. During *training*, trajectories are scored under GRPO with rewards combining (i) code correctness, (ii) exploration coverage of $\mathcal{D}$, and (iii) confidence in the summary (measured by perplexity). During *inference*, we sample multiple trajectories per query. Uncertainty is estimated via retrieval entropy and CoCoA; if uncertainty is high, the agent abstains. Otherwise, the lowest-perplexity summary is returned, accompanied by confidence scores. Full algorithmic details are in Algorithm A2 Appendix A.

## 3.3 UNCERTAINTY SIGNALS

**Summary uncertainty (training: Perplexity, inference: CoCoA).** We adopt perplexity-based CoCoA from Vashurin et al. (2025), which unifies two signals: token-level confidence (perplexity) and semantic consistency across samples. The resulting Minimum-Bayes-Risk-derived score $u_{\text{CoCoA}}$ aligns more strongly with true error rates than either component alone. At inference, we sample $K$ candidate summaries and compute CoCoA to accept or abstain. By construction, CoCoA already integrates perplexity, so no separate perplexity term is calculated at inference; during training, we use perplexity $u_{\text{Perp}}$ alone as a cheaper proxy. Full details are described in Appendix A.

Example (CoCoA). For a query on "biomarkers associated with survival in cancer patients", one episode yields "The upregulation of genes X, Y, and Z is associated with a significant decrease in predicted survival time for patients with aggressive cancer types," while another outputs "The expression levels of genes X, Y, and Z show no correlation with survival outcomes across the patient cohort." Low cross-sample consistency raises the CoCoA score, signaling semantic inconsistency and triggering abstention despite both trajectories being individually plausible.

**Retrieval uncertainty (inference-only).** High-dimensional databases pose challenges in table selection; we address this by quantifying retrieval uncertainty. For a fixed task $q$, run $K$ retrieval episodes. Let $R^{(k)}$ be the set of tables touched in episode $k$, and define the candidate set $C = \bigcup_{k=1}^{K} R^{(k)}$. The empirical selection frequency for $t \in C$ is $\hat{p}_t = \frac{1}{K} \sum_{k=1}^{K} \mathbf{1}[t \in R^{(k)}]$. We compute normalized binary entropy $H(t) = -\frac{\hat{p}_t \log \hat{p}_t + (1 - \hat{p}_t) \log(1 - \hat{p}_t)}{\log 2}$ and aggregate

$$u_{\text{ret}}(q) = \frac{1}{|C|} \sum_{t \in C} H(t). \tag{1}$$

High $u_{\text{ret}}$ indicates inconsistent evidence acquisition. We compute $u_{\text{ret}}$ during inference but omit it as a training reward due to the high computational cost of sampling.

Example (Retrieval Uncertainty). For query "biomarkers associated with survival in cancer X", the agent first invokes `SQLExecutor` to retrieve candidate gene–expression tables. It then issues a second targeted SQL to join clinical survival labels. If repeated episodes select different tables, retrieval uncertainty $u_{\text{ret}}$ is high, indicating unstable evidence and triggering abstention at inference.

### 3.4 Training rewards

We use three terminal reward components: (i) *Code execution* which rewards the agent for correctly executing SQL queries and Python code, teaching it to effectively navigate the environment; (ii) an *LLM-judge* score, which promotes broad, grounded factual coverage, encouraging exploration of the dataset environment for information; and (iii) *summary confidence*, which favors low-uncertainty summaries, promoting the exploitation of existing knowledge. The reward is a weighted sum $R(\tau) = \alpha_{\text{code}} R_{\text{code}} + \alpha_{\text{judge}} R_{\text{judge}} + \alpha_{\text{conf}} R_{\text{conf}}$. Formulas and weights are given in Appendix A.

**Schedules.** To balance exploration ($R_{\text{Judge}}$) and exploitation ($R_{\text{conf}}$) over the 100 training steps $t$, we make $\alpha_{\text{conf}}$ depend on $t$ and introduce reward schedules. The *Baseline* schedule ($R \equiv R_{base}$) applies fixed weights throughout training but risks harming early exploration of the dataset. *Two-Phase* ($R \equiv R_{phase}$) prioritizes exploration in early steps and adds exploitation midway through. *Stepwise Addition* ($R \equiv R_{step}$) periodically boosts $R_{\text{conf}}$ at regular intervals, while retaining exploration focus. *Adaptive Exploitation* ($R \equiv R_{adapt}$) dynamically adjusts $\alpha_{\text{conf}}$ based on intermediate $R_{\text{Judge}}$ performance, integrating continuous exploitation that gradually tapers off as summaries stabilize. See Table A4 in Appendix A for details.

### 3.5 Optimization with GRPO

We train with *Group Relative Policy Optimization* (GRPO), a PPO-style objective with a KL penalty to a reference policy $\pi_{ref}$, effective for reasoning LLMs (Shao et al., 2024; Guo et al., 2025; Liu et al., 2025b; Singh et al., 2025). With ratio $r_\theta(\tau) = \pi_\theta(\tau)/\pi_{\text{old}}(\tau)$, we maximize

$$\mathcal{L}(\theta) = \mathbb{E}_\tau \Big[ \min \big( r_\theta A, \ \text{clip}(r_\theta, 1 - \epsilon, 1 + \epsilon) A \big) \Big] - \beta \, D_{\text{KL}}(\pi_\theta \| \pi_{\text{ref}}), \tag{2}$$

where $A \equiv A(\tau)$ is the advatage of trajectory $\tau$, derived from the reward $R(\tau)$.

### 3.6 Inference: Post-output filtering

At inference we sample $K$ trajectories, compute $u_{\text{ret}}$ and $u_{\text{CoCoA}}$, and apply a conservative rule: abstain if the sum exceeds a tuned threshold $2\kappa$; otherwise emit the candidate with lowest $u_{\text{Perp}}$ and use $u_{\text{ret}}$ and $u_{\text{CoCoA}}$ as reliabilty scores. Threshold values are determined on a validation split through human inspection. Details are described in algorithm A2 in Appendix A.

## 4 Experiments

### 4.1 Datasets

We evaluate our approach on two multi-omics databases: one public benchmark and one internal, proprietary dataset. The MLOmics benchmark, which focuses on cancer research, has a flat structure with only 45 tables, and consists mostly of raw measurements, while the internal dataset features a tree-like schema with over 2,000 tables and includes aggregated summary statistics. This diversity allows us to assess whether our agent remains robust across (i) compact, unprocessed data scenarios, and (ii) highly structured, large-scale environments, as the schema is shown in Appendix B.

For agents, evaluating across multiple environments is critical: policies often overfit to the dynamics of a single environment schema and fail to generalize when the relational structure or data granularity changes Subbaswamy et al. (2021); Jiang et al. (2023). Recent work on environment generalization in RL Gu et al. (2025); Teoh et al. (2025) shows that agents trained in one setting may exploit spurious regularities and collapse when exposed to even minor distributional shifts. In line with these findings, we deliberately test on both a compact raw benchmark and a large schema-rich dataset to probe whether our approach adapts robustly to environment variation.

**MLOmics dataset.** We also evaluate on the MLOmics benchmark Yang et al. (2025), an open cancer multi-omics dataset with 8,314 patient samples across 32 cancer types. It provides four modalities—mRNA, microRNA, DNA methylation, and copy number variation. We use the *Top* feature version (ANOVA-selected subsets), which offers a standardized and reproducible public testbed complementing our internal dataset. Details and visuals of the dataset schemas are available in Appendix B.

**Internal multi-omics dataset.** Our internal dataset stems from layered biomedical omics. While the contents are proprietary, it includes tens to thousands of tables across transcriptomics, proteomics, and metabolomics. The schema combines a tree-like hierarchy from root entities with a broad relational structure hinging on a central table—making it a compelling testbed for agent adaptability.

## 4.2 IMPLEMENTATION DETAILS

The datasets are split into training and testing sets with a 70:30 ratio based on patient samples (Figure 1), ensuring consistent representation of all tables. We define 100 summary tasks per dataset, validated by scientists (examples in Appendix C), evaluated by LLMs and domain experts, and designed to capture the most relevant information comprehensively. Of these, 80 tasks are used for training and 20 for evaluation. During inference, each task is answered five times, and we report the mean and standard deviation of the scores for robustness.

All experiments utilize the `ART` framework[1], with `Qwen2.5-14B-Instruct` employed as the policy backbone. Training is conducted on a single NVIDIA A100 GPU. Hyperparameters are discussed in greater detail in Appendix D. Each training episode allows for up to six tool calls prior to committing a summary. During inference, $K = 5$ episodes are sampled per task to estimate retrieval and summary uncertainty.

## 4.3 METRICS

We evaluate the *quality and uncertainty of summaries* and the *reliability of uncertainty measures* as follows:

**Summary Quality.** To quantify summary quality, we report three metrics: (Q1) the total number of claims, reflecting the summary's richness in terms of content; (Q2) the ratio of correct claims, which measures factuality; and (Q3) the ratio of useful claims, which captures their relevance to the task. We derive these metrics by decomposing the summary into claims that can be assessed by an LLM fact-checking judge, following evidence that LLM judges provide reliable and fine-grained evaluations Xie et al. (2025); Zhou et al. (2025). Specifically, given a summary $s$, a task $q$, and a database $\mathcal{D}$, an o4 mini judge decomposes $s$ into atomic claims, validates them against $\mathcal{D}$ using a set of five task-specific workflows (designed in collaboration with domain experts), and assigns correctness and utility labels to each claim.

**Uncertainty.** To evaluate the model's confidence in its generated summaries, we compute the average values of $u_{\text{CoCoA}}$ (Q4) and $u_{\text{ret}}$ (Q5). To ensure this confidence is meaningful, we assess whether uncertainty estimates align with summary quality, measured by the proportion of correct claims.

Follow prior work Vashurin et al. (2025), we quantify this alignment via the Prediction Rejection Ratio (PRR):

$$\text{PRR} = \frac{\text{AUC}_{\text{unc}} - \text{AUC}_{\text{rnd}}}{\text{AUC}_{\text{oracle}} - \text{AUC}_{\text{rnd}}},$$

where $\text{AUC}_{\text{unc}}$ is obtained via uncertainty-based rejection, $\text{AUC}_{\text{rnd}}$ is a random baseline, and $\text{AUC}_{\text{oracle}}$ is an ideal oracle. Higher PRR values reflect better alignment between uncertainty and factual accuracy.

## 4.4 BASELINES

We perform a comparative analysis against a carefully chosen set of baselines that represent (a) conventional table agents, (b) reasoning agents with tool-use, (c) inference-time uncertainty filters. We

---

[1] https://art.openpipe.ai/

Table 1: Cancer Multi-Omics dataset performance: average claims (Q1), correct claims (Q2), and useful claims (Q3) per summary, with correctness/usefulness ratios; we also report uncertainty metrics $u_{\text{CoCoA}}$ (Q4) and $u_{\text{ret}}$ (Q5); for each, the value outside parentheses is the uncertainty ($\downarrow$), and the value in parentheses is PRR ($\uparrow$); arrows in headers indicate the direction of better results; the LangChain/ReAct agents do not produce uncertainty metrics (shown as −).

| System | # claims / summary $\uparrow$ | # correct / summary (ratio) $\uparrow$ | # useful / summary (ratio) $\uparrow$ | $u_{\text{CoCoA}} \downarrow$ (PRR $\uparrow$) | $u_{\text{ret}} \downarrow$ (PRR $\uparrow$) |
|---|---|---|---|---|---|
| LangChain agent | $5.4 \pm 0.7$ | $3.6 \pm 0.6\ (0.67 \pm 0.04)$ | $2.0 \pm 0.5\ (0.37 \pm 0.03)$ | − | − |
| ReAct agent | $5.5 \pm 0.8$ | $3.7 \pm 0.7\ (0.67 \pm 0.04)$ | $2.1 \pm 0.6\ (0.38 \pm 0.04)$ | − | − |
| Post-hoc filtering | $6.1 \pm 0.6$ | $4.2 \pm 0.5\ (0.68 \pm 0.03)$ | $2.6 \pm 0.8\ (0.43 \pm 0.03)$ | $0.40 \pm 0.05\ (0.37 \pm 0.08)$ | $0.69 \pm 0.05\ (0.25 \pm 0.07)$ |
| Ours (before training) | $2.4 \pm 0.5$ | $1.5 \pm 0.4\ (0.63 \pm 0.03)$ | $0.9 \pm 0.3\ (0.40 \pm 0.04)$ | $0.47 \pm 0.05\ (0.37 \pm 0.09)$ | $0.84 \pm 0.06\ (0.24 \pm 0.07)$ |
| Ours ($R_{adapt}$, before filtering) | $10.2 \pm 1.3$ | $8.4 \pm 1.1\ (0.82 \pm 0.03)$ | $4.0 \pm 0.8\ (0.39 \pm 0.04)$ | $0.25 \pm 0.04\ (0.38 \pm 0.09)$ | $0.67 \pm 0.05\ (0.25 \pm 0.06)$ |
| Ours ($R_{adapt}$, after filtering) | $10.5 \pm 1.5$ | $9.9 \pm 1.2\ (0.94 \pm 0.02)$ | $4.5 \pm 0.9\ (0.43 \pm 0.03)$ | $0.19 \pm 0.03\ (0.45 \pm 0.08)$ | $0.44 \pm 0.04\ (0.28 \pm 0.08)$ |

include (i) a standard LangChain SQL agent[2] augmented with Python-based tools. This agent translates natural-language questions into SQL, executes the query, and produces one-shot summaries without any uncertainty modeling. We use `OpenAI-o4-mini` as the backbone, a strong model for agents and structured database tasks. (ii) a ReAct-style agent that interleaves reasoning traces with SQL/Python tool calls but, again, using `OpenAI-o4-mini` as the backbone; (iii) a post-hoc filtering baseline that uses an untrained model and applies CoCoA thresholds after generation, isolating the value of inference-time abstention from uncertainty-aware training. (iv) our agent before GRPO training; (v) our model after GRPO training, which incorporates uncertainty-aware reward shaping; and (vi) our GRPO-trained agent with inference-time filtering as described in Section 3.6.

## 4.5 RESULTS

Our uncertainty-aware agent advances multi-table summarization, delivering significant improvements in summary quality and reliability across both test datasets, as evidenced in Tables 1 and 2. As the first to tackle this task with the MLOmics dataset, our approach sets a new benchmark, producing more claims with substantially higher correctness and usefulness ratios. Correctness increased from 1.5 to 9.9 average correct claims per summary in the cancer multi-omics dataset and from 0.9 to 8.4 in the internal dataset, a clear demonstration of the power of uncertainty-based rewards in curbing spurious outputs. Usefulness ratios rose from 0.60 to 0.78 on the internal dataset, reflecting enhanced schema navigation and evidence synthesis across diverse environments.

These gains generalize across a proprietary schema-rich multi-omics corpus and the MLOmics benchmark, underscoring the agent's adaptability. While the lack of prior work on this specific task/dataset combination highlights the pioneering nature of our results, they also outstrip the LangChain SQL-agent baseline (e.g., 3.6 vs. 9.9 correct claims in cancer, 3.0 vs. 8.4 internally) and the ReAct Agent, which lack uncertainty modeling. Our approach sharpens uncertainty estimates, with retrieval entropy ($u_{\text{ret}}$) and summary uncertainty ($u_{\text{CoCoA}}$) decreasing, signaling more stable evidence acquisition and consistent outputs. The Prediction Rejection Ratio (PRR) improvements—rising to 0.45 (cancer) and 0.47 (internal) for CoCoA—validate that uncertainty signals serve as potent control mechanisms, aligning confidence with factual reliability and enhancing trustworthiness.

Focusing on the filtering step during inference, this component improved performance metrics, boosting correctness from 0.82 to 0.94 (cancer) and from 0.84 to 0.90 (internal), while usefulness climbed from 0.39 to 0.43 and 0.71 to 0.78, respectively. This underscores the critical role of inference-time refinement in producing reliable summaries across heterogeneous settings.

## 4.6 ABLATION

We study four factors that could explain our improvements: reward schedules, uncertainty signals, judge dependence, and inference thresholds. Tables and details can be found in Appendix E.

**Reward schedules.** Reward shaping substantially affects optimization trajectories. Table A5 compares $R_{zero}$, $R_{base}$, $R_{phase}$, $R_{step}$, and $R_{adapt}$. Importantly, we include **RZero**, an RL baseline trained without any uncertainty-derived rewards, to demonstrate that gains are not merely due to

---

[2]https://python.langchain.com/docs/integrations/tools/sql_database/

Table 2: Internal dataset performance: average claims (Q1), correct claims (Q2), and useful claims (Q3) per summary, with correctness/usefulness ratios; we also report uncertainty metrics $u_{CoCoA}$ (Q4) and $u_{ret}$ (Q5); for each, the value outside parentheses is the uncertainty ($\downarrow$), and the value in parentheses is PRR ($\uparrow$); arrows in headers indicate the direction of better results; the LLangChain/ReAct agents do not produce uncertainty metrics (shown as –).

| System | # claims / summary $\uparrow$ | # correct / summary (ratio) $\uparrow$ | # useful / summary (ratio) $\uparrow$ | $u_{CoCoA} \downarrow$ (PRR $\uparrow$) | $u_{ret} \downarrow$ (PRR $\uparrow$) |
|---|---|---|---|---|---|
| LangChain agent | $4.5 \pm 0.8$ | $3.0 \pm 0.6\,(0.67 \pm 0.05)$ | $3.0 \pm 0.5\,(0.65 \pm 0.04)$ | – | – |
| ReAct agent | $4.5 \pm 0.7$ | $2.9 \pm 0.5\,(0.64 \pm 0.04)$ | $2.9 \pm 0.6\,(0.64 \pm 0.03)$ | – | – |
| Post-hoc filtering | $5.0 \pm 1.2$ | $3.4 \pm 0.9\,(0.68 \pm 0.03)$ | $3.3 \pm 0.8\,(0.66 \pm 0.03)$ | $0.36 \pm 0.05\,(0.38 \pm 0.07)$ | $0.76 \pm 0.06\,(0.30 \pm 0.06)$ |
| Ours (before training) | $1.5 \pm 0.3$ | $0.9 \pm 0.2\,(0.60 \pm 0.03)$ | $0.9 \pm 0.2\,(0.60 \pm 0.02)$ | $0.45 \pm 0.05\,(0.39 \pm 0.09)$ | $0.84 \pm 0.04\,(0.29 \pm 0.07)$ |
| Ours ($R_{adapt}$, before filtering) | $9.3 \pm 1.2$ | $7.2 \pm 1.0\,(0.84 \pm 0.03)$ | $6.6 \pm 0.7\,(0.71 \pm 0.04)$ | $0.27 \pm 0.04\,(0.42 \pm 0.08)$ | $0.65 \pm 0.07\,(0.33 \pm 0.07)$ |
| Ours ($R_{adapt}$, after filtering) | $9.3 \pm 1.1$ | $8.4 \pm 0.9\,(0.90 \pm 0.02)$ | $7.2 \pm 0.8\,(0.78 \pm 0.03)$ | $0.20 \pm 0.04\,(0.47 \pm 0.08)$ | $0.42 \pm 0.06\,(0.38 \pm 0.08)$ |

Table 3: Concordance index (c-index) scores on the held-out test set: results compare a LangChain baseline with our method under different refinement strategies ($R_{base}$, $R_{phase}$, $R_{step}$, $R_{adapt}$); higher values indicate better predictive alignment.

| Model | LangChain/ReAct agent | Ours (before training) | Ours ($R_{base}$) | Ours ($R_{phase}$) | Ours ($R_{step}$) | Ours ($R_{adapt}$) |
|---|---|---|---|---|---|---|
| C-index | 0.22/0.21 | 0.32 | 0.55 | 0.60 | 0.64 | 0.63 |

reinforcement learning itself but arise from incorporating uncertainty as a control signal. $R_{adapt}$ yields the highest useful-claims ratio (0.78 vs 0.30 for $R_{base}$ on Internal) and stronger PRR alignment. Learning curves (Fig. A5) show $R_{adapt}$ avoids early collapse seen in $R_{base}$, indicating that adaptive weighting of uncertainty stabilizes training.

**Uncertainty signals.** We ablate the contributions of individual uncertainty signals within the $R_{judge}$ schedule by training with each signal in isolation. Specifically, we compare summary-based uncertainties vs retrieval-based uncertainties and consistency based uncertainties vs information theoretic ones. Perplexity yields a baseline Useful Ratio of 0.78 with PRR 0.47. CoCoA improves calibration slightly (Useful Ratio 0.72, PRR 0.50) but requires $2.6\times$ more compute and adapts more slowly. Entropy (0.76, PRR 0.46) and retrieval variance (0.76, PRR 0.39) achieve stronger cost–benefit tradeoffs. The mechanism seems to be: training purely for consistency encourages rigidity, while lighter signals adapt more flexibly. Full results are in Appendix Table A6.

**Judge robustness.** Optimizing a single judge invites reward hacking (Ziegler et al., 2019; Gao et al., 2023). We compared $R_{adapt}$ models scored by (i) our strong $R_{judge}$, (ii) a weaker LLM judge (GPT-4.1 Nano and Gemini 2.5 Flash Lite), and (iii) a 40-query human holdout. Correlations were moderate-to-strong ($r = 0.62 \pm 0.08$ vs weak; $r = 0.64 \pm 0.07$ vs human). Importantly, *system rankings were identical*: $R_{adapt} > R_{step}¿, R_{phase} > R_{base}$. This indicates the gains are not artifacts of one evaluator.

**Inference thresholds.** Finally, we varied uncertainty thresholds $\kappa \in \{0.2, 0.5, 0.8\}$ for post-hoc filtering. Table A8 shows the trade-off: higher $\kappa$ reduces coverage but improves precision. $R_{adapt}$ models dominate at all thresholds.

## 4.7 PREDICTION RESULTS

Beyond evaluating summary correctness and usefulness, it is important to test whether the agent's knowledge transfer produces meaningful downstream outcomes. Survival prediction provides such a test, connecting textual reasoning with a clinically relevant endpoint. To perform this task, we prompt the agent to estimate survival times for held-out patients by leveraging in-context knowledge from summaries related to survival, rather than task-specific supervision. The predictions are evaluated using the concordance index (C-index), which measures how well predicted survival times align with ground-truth outcomes.

As shown in Table 3, our framework consistently outperforms the LangChain baseline, with the largest improvement from $R_{step}$. The stable performance highlights the role of uncertainty-aware refinement in producing reliable predictions.

Additionally, it is important to note that untrained agents performed worse than random chance. They exhibited a tendency to systematically focus on incorrect features drawn from the literature, rather than accurately interpreting the dataset. This underscores the necessity of training and appropriate methodology to improve predictive performance.

## 5 DISCUSSION

We propose uncertainty-aware LLM agents that integrate retrieval and summary uncertainty into training and inference, addressing the challenge of reliable tabular summarization. The key contribution is to treat uncertainty as a **control signal** that shapes optimization, guides agent behavior, and governs inference-time filtering.

Empirically, uncertainty-aware agents produce nearly twice as many useful claims as baseline SQL agents, with gains confirmed by fact-checking and downstream survival analysis. Uncertainty estimates themselves are predictive: the PRR roughly doubles, showing that confidence tracks factual reliability. Thus, uncertainty serves as an **actionable lever** rather than a diagnostic byproduct. A central design principle emerges: **agents should know when not to answer**. Abstention on high-uncertainty queries and filtering of synthetic data yield a conservative, safety-first behavior, crucial in biomedical applications but relevant more broadly.

Although evaluated on biomedical multi-omics data, the framework is domain-agnostic and applies to finance, e-commerce, or clinical EHRs. Its components—entropy-based retrieval uncertainty, self-consistency signals, and GRPO training—are modular and can be integrated into existing pipelines without architectural changes. Embedding uncertainty into the **decision loop** does not eliminate hallucinations but enables calibrated, trustworthy behavior beyond post-hoc filtering, moving toward safer deployment in high-stakes settings.

## 6 LIMITATIONS

Our current evaluation is limited to biomedical multi-omics data. While this domain highlights the need for reliability in high-stakes settings, testing across finance, e-commerce, and other structured environments will demonstrate the broader generality of the framework.

We also rely on automated LLM-based judges for reward shaping and fact-checking. This enables scalable experimentation but could potentially induce bias. Expanding systematic human validation will be an important next step, and our uncertainty annotations can help guide such expert audits.

Finally, the method requires multiple rollouts (e.g., K=5) and CoCoA-based self-consistency, which add inference cost. Preliminary results suggest smaller K retains most benefits, and leveraging a lightweight uncertainty proxy eg. perplexity instead of CoCoA could make the approach more efficient.

## 7 CONCLUSION

This work shows that uncertainty can be treated not just as a diagnostic signal, but as an active control mechanism for agentic systems operating over structured data. By combining retrieval and summary uncertainty during both training and inference, our agent learns when to proceed and when to abstain, improving both correctness and safety in multi-table reasoning tasks. While early results suggest benefits for downstream analysis, open challenges remain in calibration, evaluation beyond proprietary datasets, and reducing inference costs. We see this as a step toward building agents that scale responsibly, and we invite the community to explore stronger uncertainty estimation methods, richer benchmarks, and ethical safeguards.

## 8 ETHICS

This work relies primarily on publicly available, de-identified datasets (e.g., MLOmics and other open cancer databases) under their original licenses; no patient-identifiable data were used. Code, prompts, and configurations will be released to support replication, with hyperparameters and training details in Appendix D. Automated judge scores were validated against expert assessments (N=40), with agreement metrics and audit protocols in the supplementary materials. The system is intended for research purposes only and should not be considered medical advice without expert validation.

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

## A   ADDITIONAL METHODS DETAILS

This section collects additional details about our setup that were omitted from the main text for clarity.

**Summary Uncertainty.**   *Perplexity.* For a summary token sequence $s_{1:T}$:

$$u_{\text{Perp}}(s_{1:T}) = \exp\left(-\tfrac{1}{T}\sum_{t=1}^{T}\log p_\theta(s_t \mid s_{<t}, \text{context})\right), \tag{3}$$

where $p_\theta(s_t \mid s_{<t}, \text{context})$ represents the probability assigned by the model to token $s_t$ given the sequence of preceding tokens $s_{<t}$ and any task-specific contextual information. Lower perplexity implies higher model confidence in the token-level generation process.

*CoCoA.* We use the COCOA metric (Vashurin et al., 2025), which enhances perplexity-based confidence – relying solely on LLM probabilities and providing no information about the answer distribution – with semantic self-consistency.

Given an actual output sequence $s^*$ and $K-1$ sampled sequences $s^{(k)}, k = 1, ..., K-1$, we compute a consistency-based uncertainty metric Vashurin et al. (2025)

$$u_{\text{cons}}(s^*, \{s^{(k)}\}) = 1 - \frac{1}{K-1}\sum_{k=1}^{K-1}\text{sim}(s^{(k)}, s^*),$$

where, for the similarity metric sim we use the RoBERTa-large cross-encoder model, fine-tuned on the Semantic Textual Similarity benchmark dataset Reimers & Gurevych (2019); Liu et al. (2019); Cer et al. (2017). Multiplying $u_{\text{cons}}(s^*, \{s^{(k)}\})$ with the perplexity of $s^*$ produces the CoCoA metric $u_{\text{CoCoA}}(s^*, \{s^{(k)}\})$.

**Reward Design.** *Code Execution Reward.* To incentivize correct database interactions, the trajectory receives a reward based on the number of correctly executed SQL queries or Python code executions, with a stronger emphasis on rewarding initial successes to encourage learning. Let $x(\tau) \in \mathbb{N}$ be the number of correctly executed code actions in trajectory $\tau$. The code execution reward is:

$$R_{code}(\tau) = \min\left(1, \frac{\log(10x(\tau) + 1)}{\log(31)}\right),$$

where the reward is capped at a maximum value of 1 for three correctly executed actions. This design aims to teach the model to produce correct executable code early in training. The reward cap ensures the model saturates the benefit from code execution once it reliably achieves three successful actions, encouraging it to focus on higher-level tasks, such as summary generation, as training progresses.

*Exploration Judge Reward.* An external `o4-mini` LLM counts the number $c(\tau)$ of grounded, non-overlapping atomic facts in the trajectory $\tau$ that are relevant to the user's topic. The reward is:

$$R_{\text{Judge}}(\tau) = \min\left(\frac{c(\tau)}{20}, 1\right),$$

promoting thorough database exploration to uncover relevant and diverse information. The normalization factor 20 reflects our empirical observation that trajectories with around 20 grounded, non-overlapping facts typically provide sufficient diversity and coverage for most queries.

*Summary Confidence Reward.* The inverse perplexity of the generated summary $s(\tau)$ corresponding to the trajectory $\tau$, serves as a measure of token-level confidence, normalized to $(0, 1]$:

$$R_{\text{conf}}(\tau) = \frac{1}{u_{\text{Perp}}(s(\tau))}.$$

While $R_{\text{Judge}}$ promotes database exploration, $R_{\text{conf}}$ incentivizes exploitation by rewarding low-uncertainty summaries. Consistency-based uncertainty metrics, such as CoCoA, are omitted during training to sidestep the high computational overhead of sampling.

**Reward Schedules.** We explore various reward schedules over the 100 training steps $t$ to balance exploration and exploitation. A summary of these schedules is provided in Table A4. Constants are empirically chosen to balance the contributions of individual reward components, ensuring effective training dynamics. Ablation studies of these constants are left for future work.

Table A4: Summary of reward schedules, their formulas, and descriptions.

| Schedule | Formula | Description |
|---|---|---|
| Zero | $R_{zero}(\tau) = R_{code}(\tau) + 4R_{Judge}(\tau)$ | Does not use the uncertainty signal in reward. |
| Baseline | $R_{base}(\tau) = R_{code}(\tau) + 4R_{Judge}(\tau) + \frac{1}{3}R_{\text{conf}}(\tau)$ | Uses a fixed combination of all three reward components. |
| Two-Phase | $R_{phase}(\tau) = \begin{cases} R_{code}(\tau) + 4R_{Judge}(\tau), & \text{if } t \leq 50, \\ R_{code}(\tau) + 4R_{Judge}(\tau) + \frac{1}{3}R_{\text{conf}}(\tau), & \text{if } t > 50. \end{cases}$ | Focuses on exploration during the first half, incorporates exploitation in the second training half. |
| Stepwise Addition | $R_{step}(\tau) = \begin{cases} R_{code}(\tau) + 4R_{Judge}(\tau), & \text{if } t \bmod 10 \neq 0, \\ R_{code}(\tau) + 4R_{Judge}(\tau) + 2R_{\text{conf}}(\tau), & \text{if } t \bmod 10 = 0. \end{cases}$ | Periodically emphasizes exploitation every 10 steps. |
| Adaptive Exploitation | $\alpha = \exp\left(-50\left(R_{\text{Judge}}(\tau) - \frac{1}{2}\right)^2\right),$ $R_{adapt}(\tau) = R_{code}(\tau) + 4R_{Judge}(\tau) + 2\alpha R_{\text{conf}}(\tau).$ | Initially promotes exploration, then gradually integrates exploitation, and tapers off to prevent generic summaries. |

**Episode algorithms.** Figure A2 describes algorithms for full training and inference episodes.

---

**Algorithm 1** Episode Algorithms for Training and Inference

---

  **procedure** SINGLE_EPISODE($q, \mathcal{D}, \pi_\theta$, M)
    Initialize empty trajectory $\tau \leftarrow \{\}$
    **for** $t = 1, ..., $ M **do**
      Sample action $a_t \sim \pi_\theta(\cdot \mid x_t)$
      **if** $a_t$ is `SQLExecutor(query)` **then**
        $r_t \leftarrow$ Execute SQL query on $\mathcal{D}$
        Append $(a_t, r_t)$ to $\tau$
      **else if** $a_t$ is `PythonTool(code)` **then**
        $r_t \leftarrow$ Execute Python code on relevant database parts
        Append $(a_t, r_t)$ to $\tau$
      **else if** $a_t$ is `Schema(Table)` **then**
        $r_t \leftarrow$ Retrieve schema of the specified table
        Append $(a_t, r_t)$ to $\tau$
      **else if** $a_t$ is `CommitSummary(summary)` **then**
        Extract summary $s$, including token logits
        Append $(a_t, s)$ to $\tau$
        **break**
      **end if**
    **end for**
    **return** $\tau, s$
  **end procedure**
  **procedure** TRAINING_EPISODE($q, \mathcal{D}, \pi_\theta$, M)
    $\tau, s \leftarrow$ SINGLE_EPISODE($q, \mathcal{D}, \pi_\theta, M$)
    Compute token-level perplexity $u_{PPL}(s)$
    Compute rewards $R_{Judge}(\tau), R_{code}(\tau), R_{conf}(\tau)$
    Combine $R_{Judge}(\tau), R_{code}(\tau), R_{conf}(\tau)$ to compute terminal reward $R(\tau)$
    Store $(\tau, R(\tau))$ for GRPO update
  **end procedure**
  **procedure** INFERENCE_EPISODE($q, \mathcal{D}, \pi_\theta$, K)
    Initialize $\mathcal{S} \leftarrow \{\}$ and $\mathcal{T} \leftarrow \{\}$
    **for** $k = 1, ..., K$ **do**
      $\tau_k, s_k \leftarrow$ SINGLE_EPISODE($q, \mathcal{D}, \pi_\theta, M$)
      Append $s_k$ to $\mathcal{S}$, $\tau_k$ to $\mathcal{T}$
    **end for**
    Compute summary uncertainty $u_{\text{CoCoA}}(\mathcal{S})$
    Compute retrieval uncertainty $u_{\text{ret}}(\mathcal{T})$ from SQL queries in trajectories
    $\tilde{\tau}, \tilde{s} \leftarrow$ (trajectory, summary) pair with lowest-perplexity summary from zip($\mathcal{T}, \mathcal{S}$)
    Store $(\tilde{\tau}, \tilde{s}, u_{\text{CoCoA}}(\mathcal{S}), u_{\text{ret}}(\mathcal{T}))$
  **end procedure**

---

Figure A2: Episode algorithms. Training uses a single episode to compute terminal reward based on code execution, confidence and exploration. Inference samples multiple episodes to compute summary and retrieval uncertainties.

# B DATASETS

This section describes the datasets used in our experiments.

## B.1 INTERNAL MULTI-OMICS DATASET

The internal dataset is built from multi-layered omics data. While the specific table contents cannot be disclosed, its structure can be summarized as:

- **Architecture:** Multi-layered, with each layer corresponding to a distinct omics modality (e.g., transcriptomics, proteomics, metabolomics).

- **Scale:** Each layer consists of between tens and hundreds of relational tables.
- **Topologies:** Two primary schema structures are observed: (a) a *tree-like hierarchy*, in which child tables branch recursively from root entities, and (b) a *broad schema*, in which many tables connect directly to a central entity.

These schema variations provide structurally distinct environments that stress-test an agent's ability to adapt to different database organizations.

## B.2 DATASET SCHEMATIC FOR INTERNAL MULTI-OMICS DATASET

Figure A3 contains dataset Schematic for the Internal Multi-Omics Dataset.

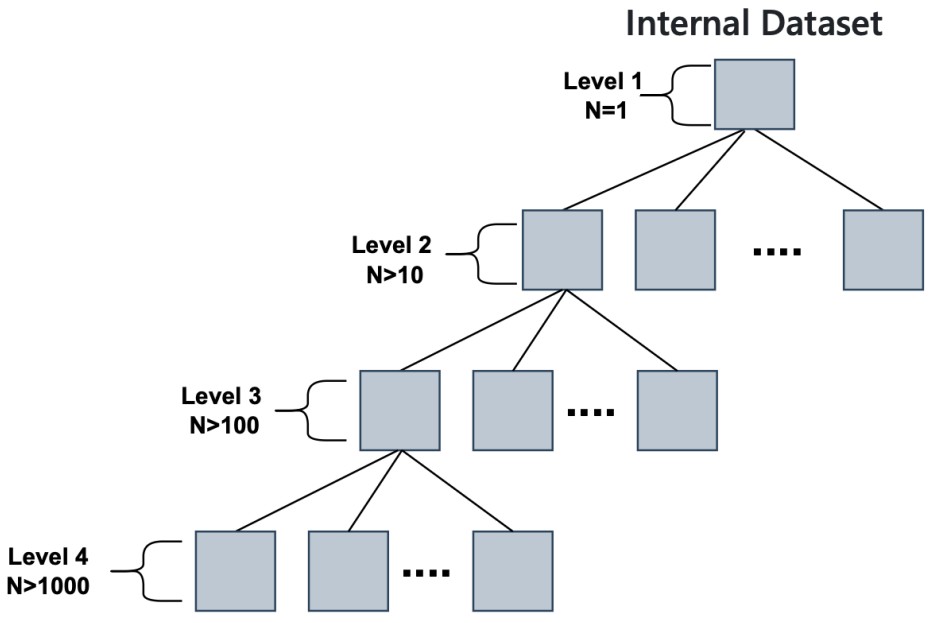

Figure A3: Internal multi-omics dataset showing tree-like schema topology.

## B.3 MLOMICS: CANCER MULTI-OMICS DATABASE FOR MACHINE LEARNING

MLOmics Yang et al. (2025) is an open multi-omics dataset comprising 8,314 patients across 32 cancer types. It provides four standardized omics modalities:

- **mRNA expression:** Gene-level transcriptional profiles.
- **microRNA expression:** Small noncoding RNAs regulating gene expression.
- **DNA methylation:** CpG site methylation fractions representing epigenetic regulation.
- **Copy number variation (CNV):** Segment-level gene copy alterations.

Each modality is released in three feature versions:

- *Original*: full feature set,
- *Aligned*: subsets harmonized across modalities,

- *Top*: statistically filtered subsets (ANOVA-based).

MLOmics additionally includes baseline machine learning benchmarks (6–10 methods), clustering and survival analyses, and external knowledge integration (STRING, KEGG). These resources make it a reproducible benchmark for developing and evaluating uncertainty-aware agents.

### B.4 DATASET SCHEMATIC FOR MLOMICS

Figure A4 contains dataset Schematic for Cancer MLOmics Dataset.

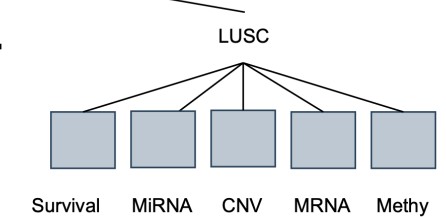

Figure A4: Public MLOmics dataset, with standardized parallel modalities spanning 9 cancer types.

## C SUMMARY TASKS

This section provides examples task templates used in training and inference for the Cancer MLOmics Dataset. Each task outlines specific objectives and details the steps required to obtain relevant information about different cancer types using molecular data. The complete list of tasks will be released on GitHub upon completion.

---

**Task 1: Basic Cancer-Survival Characterization**

**Objective:** For a specified cancer type CANCER_TYPE, answer the following questions:

1. How many patients are in the training set?
2. What is the median survival time?
3. What is the event rate (percentage of deaths)?
4. Describe the survival distribution.
5. Compare this cancer's survival patterns to other cancers in the database.

---

**Task 2: Molecular Data Profile**

**Objective:** For a specified cancer type CANCER_TYPE, analyze each omic layer:

1. Data distribution characteristics for each omic type.
2. Missing value analysis.
3. Create a molecular profile summary specific to this cancer type.

---

---

**Task 3: Cancer-Specific Biomarkers**

**Objective:** For a specified cancer type `CANCER_TYPE`, identify and analyze biomarkers:

1. Identify top survival-associated features from each omic type:
   - Top 20 mRNA features
   - Top 20 miRNA features
   - Top 20 methylation sites
   - Top 20 CNV regions
2. Analyze their biological relevance.
3. Compare with known markers for this cancer type.
4. Create a prioritized biomarker list.

---

**Task 4: Multi-omic Integration**

**Objective:** For a specified cancer type `CANCER_TYPE`, integrate various omic layers:

1. Find correlations between features across different omic types.
2. Identify multi-omic patterns associated with survival.
3. Create an integrated molecular profile.
4. Highlight unique molecular characteristics of this cancer type.

---

**Task 5: Clinical-Molecular Summary**

**Objective:** Create a comprehensive summary for a specified cancer type `CANCER_TYPE`:

1. Key survival characteristics.
2. Most important molecular features.
3. Multi-omic patterns.
4. Clinical-molecular associations.
5. Comparison with other cancer types.
6. Potential clinical implications.

---

## D    HYPERPARAMETERS

The backbone of the model used in this work is Qwen2.5-14B-Instruct, implemented within the ART framework. Training was conducted on 1×NVIDIA A100 80GB GPU, with a total computational cost of approximately $22$ GPU-hours per model. We use sampling defaults of $M = K = 5$.

The training process employs Grouped Relative Policy Optimization (GRPO) to optimize the summarization agent. We set the clipping parameter $\epsilon = 0.2$ and the KL penalty weight $\beta = 0.01$. The learning rate is defined as `5e-5`, selected after searching for optimal values in the range between `1e-7` and `1e-4`. The model is allowed up to $6$ tool calls per query for performing retrievals and summary generation, determined through a search over $4\{10$ tool calls per query, where only marginal improvements were observed beyond $6$ tool calls.

Training is conducted in mini-batches consisting of `3 groups per step`, with each group containing `4 rollouts`, ensuring that every query is processed multiple times as part of GRPO optimization. Each training run spans `4 epochs`.

All code, prompts, and configuration files will be released to ensure reproducibility.

Table A5: Reward schedule ablations across the internal and Multi-Omics cancer datasets: average number of claims per summary, claim correctness and usefulness ratios; we also report uncertainty metrics $u_{\text{CoCoA}}$ and $u_{\text{ret}}$ with PRR indicating alignment with correctness.

| Schedule | # claims | Internal Correct ratio | Useful ratio | Internal (UQ) $u_{\text{CoCoA}}$/PRR | $u_{\text{ret}}$/PRR | # claims | Cancer Correct ratio | Useful ratio | Cancer (UQ) $u_{\text{CoCoA}}$/PRR | $u_{\text{ret}}$/PRR |
|---|---|---|---|---|---|---|---|---|---|---|
| $R_{zero}$ | $5.2 \pm 0.3$ | $0.27 \pm 0.05$ | $0.27 \pm 0.02$ | 0.51/0.35 | 0.86/0.25 | $5.5 \pm 0.4$ | $0.33 \pm 0.06$ | $0.29 \pm 0.03$ | 0.49/0.36 | 0.87/0.24 |
| $R_{base}$ | $4.5 \pm 0.4$ | $0.64 \pm 0.04$ | $0.30 \pm 0.03$ | 0.13/0.33 | 0.72/0.28 | $5.2 \pm 0.5$ | $0.65 \pm 0.04$ | $0.25 \pm 0.03$ | 0.14/0.31 | 0.75/0.26 |
| $R_{phase}$ | $6.0 \pm 0.5$ | $0.67 \pm 0.03$ | $0.50 \pm 0.03$ | 0.15/0.39 | 0.65/0.33 | $6.8 \pm 0.6$ | $0.66 \pm 0.03$ | $0.41 \pm 0.03$ | 0.17/0.34 | 0.68/0.28 |
| $R_{step}$ | $8.3 \pm 0.6$ | $0.75 \pm 0.03$ | $0.55 \pm 0.02$ | 0.22/0.32 | 0.58/0.32 | $9.0 \pm 0.7$ | $0.78 \pm 0.02$ | $\mathbf{0.44 \pm 0.03}$ | 0.25/0.39 | 0.61/0.29 |
| $R_{adapt}$ | $9.3 \pm 1.1$ | $\mathbf{0.90 \pm 0.02}$ | $\mathbf{0.78 \pm 0.03}$ | 0.20/0.47 | 0.42/0.38 | $10.5 \pm 1.5$ | $\mathbf{0.94 \pm 0.01}$ | $0.43 \pm 0.03$ | 0.19/0.45 | 0.44/0.28 |

# E   ABLATION DETAILS

## E.1   REWARD SCHEDULES

We evaluate five reward schedules ($R_{\text{zero}}$, $R_{\text{base}}$, $R_{\text{phase}}$, $R_{\text{step}}$, and $R_{\text{adapt}}$ with definitions in Table A4) to analyze the impact of uncertainty during training (Table A5). The $R_{\text{zero}}$ schedule, which excludes uncertainty rewards, has the worst correct claims ratio of 0.27 due to frequent hallucinated claims with high uncertainty.

$R_{\text{base}}$, applying uncertainty rewards from the start, improves the correct claims ratio to 0.64 but achieves limited exploration (see $R_{code}$ and $R_{Judge}$ in Figure A5), leading to shallow summaries with useful claims ratios of 0.30 for the Internal dataset and 0.25 for Cancer Multi-Omics.

To address these limitations, $R_{\text{phase}}$ defers uncertainty rewards to encourage early exploration. It raises the correct claims ratio to 0.67 and improves useful claims ratios to 0.50 on Internal and 0.41 on Cancer Multi-Omics, though outputs remain conservative and shallow due to excessive uncertainty minimization, as reflected by summary uncertainty trends in Figure A5.

$R_{\text{step}}$ introduces rewards periodically, boosting useful claims ratios to 0.55 (Internal) and 0.44 (Cancer Multi-Omics). However, abrupt uncertainty application every tenth step causes instability, reflected in unsmooth training plots in Figure A5 and inconsistent PRR values such as 0.32 for $u_{\text{CoCoA}}$ on Internal.

Finally, $R_{\text{adapt}}$ dynamically adjusts uncertainty rewards, integrating them smoothly throughout training. This yields the best performance: correct claims ratios of 0.90 and 0.94, and useful claims ratios of 0.78 (Internal) and 0.43 (Cancer Multi-Omics), with strong uncertainty alignment (e.g., PRR of 0.47 for $u_{\text{CoCoA}}$).

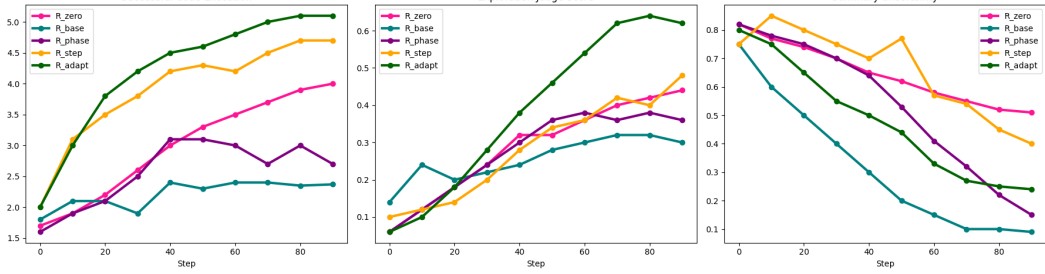

Figure A5: Performance metrics ($R_{\text{Code}}$, $R_{\text{Judge}}$, and uncertainty) during 100 training steps under different reward schedules ($R_{\text{zero}}$, $R_{\text{base}}$, $R_{\text{phase}}$, $R_{\text{step}}$, $R_{\text{adapt}}$).

## E.2   UNCERTAINTY SIGNALS

Table A6 compares four uncertainty reward signals. CoCoA improves consistency but is compute-inefficient.

Table A6: Uncertainty signal ablations (internal dataset, $R_{adapt}$ schedule).

| Signal | Useful Ratio | PRR | Relative Cost |
|---|---|---|---|
| Perplexity | $0.78 \pm 0.03$ | 0.47 | 1.0 |
| CoCoA | $0.72 \pm 0.03$ | 0.50 | 2.6 |
| Entropy | $0.76 \pm 0.02$ | 0.46 | 1.0 |
| Retrieval variance | $0.76 \pm 0.02$ | 0.39 | 2.1 |

Table A7: Judge robustness. Rankings were consistent across reward schedules (internal dataset).

| Comparison | Pearson $r$ | Ranking preserved? |
|---|---|---|
| $R_{judge}$ vs Weak Judge | $0.62 \pm 0.08$ | Yes |
| $R_{judge}$ vs Human (40q) | $0.64 \pm 0.07$ | Yes |

### E.3 JUDGE ROBUSTNESS

Table A7 gives correlations between $R_{judge}$, a weak LLM judge, and human labels. Preserved ranking: $R_{adapt} > R_{step}¿, R_{phase} > R_{base}$

### E.4 INFERENCE THRESHOLDS

Table A8 shows coverage–accuracy tradeoffs for different thresholds $\kappa$.

Table A8: Inference thresholds. Post-hoc filtering improves slightly but underperforms full uncertainty-aware training (internal dataset).

| Method | Threshold $\kappa$ | Coverage (%) | Useful Ratio | PRR |
|---|---|---|---|---|
| $R_{adapt}$ | 0.5 | 70 | $0.78 \pm 0.03$ | **0.47** |
| $R_{adapt}$ | 0.2 | 95 | $0.72 \pm 0.03$ | 0.43 |
| $R_{adapt}$ | 0.8 | 40 | $0.85 \pm 0.02$ | 0.50 |

