# OpenReview forum: "Towards Agents That Know When They Don't Know: Uncertainty as a Control Signal for Structured Reasoning"
_ICLR.cc/2026/Conference — ICLR 2026 Conference Withdrawn Submission_

### Official Review · Reviewer_8q8V · 2025-10-14

**Soundness:** 2
**Presentation:** 2
**Contribution:** 2
**Rating:** 2
**Confidence:** 5

**Summary:**

The paper tackles query-driven multi-table summarization and proposes using uncertainty as a control signal at both training and inference. During training, the authors shape rewards with perplexity/self-consistency under GRPO; at inference, they combine retrieval entropy and a summary-uncertainty score (self-consistency × LM probability) to decide pass/abstain and to filter synthetic data. On a public multi-omics dataset and a private internal dataset, the method reportedly increases correct/useful claims per summary, improves uncertainty–accuracy alignment, and raises downstream survival prediction C-index.

**Strengths:**

The idea is reasonable, but evidence is not fully convincing.

It’s unclear which component drives the gains: self-consistency, perplexity shaping, retrieval entropy, or GRPO itself. The ablations don’t isolate causal effects tightly enough.

Evaluation leans heavily on LLM-as-judge with only a small amount of human spot-checking, leaving room for judge bias or reward hacking.

The joint inference threshold is hand-tuned; there’s little analysis of sensitivity or transferability.

Compute/latency overhead from multi-rollout retrieval and self-consistency is not compared under equal budget to lighter alternatives.

**Weaknesses:**

Weak comparative baselines. Comparisons focus on conventional SQL/table agents with post-hoc filtering. Missing are stronger recent agents (e.g., modern RL/R1-style reasoning agents), end-to-end MBR/consistency calibration, or explicit UQ heads. It’s hard to credit the gains specifically to “uncertainty as a control signal.”

Over-reliance on LLM-as-judge. Human verification is limited; there’s no thorough analysis of judge agreement, drift across tasks/schemas, or defenses against reward hacking. Claim-level metrics can be systematically biased by the judge.

Thresholding sensitivity. The joint pass/abstain threshold is tuned by hand. There’s no coverage–precision trade-off curve, cross-dataset transfer, or learned-threshold variant.

Cost and efficiency. Multi-rollout retrieval plus self-consistency adds real inference cost. The paper lacks equal-budget comparisons to lighter baselines (e.g., perplexity-only or single-sample consistency) and lacks latency/throughput curves.

Reproducibility. The private multi-omics dataset carries much of the story; results on fully public tasks are too thin to support claims of domain generality. The “synthetic-data filtering” contribution lacks independent human assessment.

**Questions:**

Can you add stronger recent baselines and report significance on the main tables?

Can you provide modular ablations that isolate self-consistency, perplexity shaping, retrieval entropy, and their interactions with GRPO?

How sensitive is the joint threshold? Please show coverage–precision curves, cross-dataset transfer, and (if possible) a learned-threshold variant (e.g., utility- or risk-based selective prediction).

How do you quantify judge bias/agreement? Please report agreement across judges (strong/weak/open-source) and expand human verification with inter-annotator agreement.

Under a fixed compute budget, what is the marginal gain from CoCoA versus lighter proxies? Please include latency/throughput plots.

Can you show a small-scale result on a non-biomedical public multi-table task to back the domain-agnostic claim?

---

### Official Review · Reviewer_bXef · 2025-10-26

**Soundness:** 1
**Presentation:** 1
**Contribution:** 1
**Rating:** 2
**Confidence:** 3

**Summary:**

This work aims to develop uncertainty-aware agent for query-conditioned multi-table summarization. It proposes to combine retrieval uncertainty and summary uncertainty. Reinforcement learning with Group Relative Policy Optimization based uncertainty is used to guide inference-time filtering. Experiments were conducted on multi-omics benchmarks.

**Strengths:**

The application is interesting and important.

**Weaknesses:**

(1) The writing needs substantial improvement to make it easier to follow.
(2) Several previous works are cited in the Introduction section, but not compared in experiments.
(3) Eq (1): The retrieval uncertainty measures the uncertainty of selecting tables instead of the retrieved information. Is it possible that same information may be stored in different tables?
(4) Please elaborate how the high computational cost of calculating u_ret affect quality of services during inference.

**Questions:**

* In Algorithm 1, how to obtain the "Python code on relevant database parts"? Where does "Table" come from? The main text refer to "Algorithm 1" as "Algorithm A2"
* How to decide the reference policy pi_ref?
* Why return the candidate with the lowest u_Prep, but use u_ret and u_CoCoA as the reliabilty scores?

---

### Official Review · Reviewer_KYYx · 2025-10-30

**Soundness:** 1
**Presentation:** 1
**Contribution:** 2
**Rating:** 2
**Confidence:** 3

**Summary:**

This paper proposes an uncertainty-aware LLM agent framework for query-conditioned summarization of complex, multi-table biomedical datasets (multi-omics). The central claim is the use of "uncertainty as a control signal". The method quantifies two types of uncertainty: retrieval uncertainty (entropy over table selections during rollouts) and summary uncertainty (a combination of self-consistency and perplexity). During training, summary uncertainty (perplexity) is incorporated as a reward signal to optimize the agent using reinforcement learning (GRPO). At inference, both uncertainty signals are used to filter outputs, enabling the agent to abstain on low-confidence predictions. The authors evaluate the framework on one public and one internal multi-omics dataset, claiming significant improvements in summary factuality, calibration, and downstream survival prediction tasks.

**Strengths:**

The idea of using uncertainty, not just as a _post-hoc_ metric but as a direct control signal for RL training and inference-time abstention, is a valuable research direction and fits into the emerging area of uncertainty quantification around LLMs in expert tasks more generally, which is seeing increasing attention recently. Applying this idea to the complex domain of multi-omics analysis is well-motivated, as overconfident or incorrect summaries here, as in many biomedical tasks, can be harmful.

The paper's formulation of uncertainty into two distinct, complementary signals (retrieval/table uncertainty vs. summary/generation uncertainty) is logical and provides a clear conceptual framework.

The inclusion of a downstream survival prediction task helps evaluate the real-world utility of the method and adds depth to the analysis.

**Weaknesses:**

This core idea is unfortunately undermined by several issues in execution, experimental evaluation and the presentation of the paper itself, which leaves the central claims not very well supported.

**Statistical rigour.** The experimental methodology is not very sound. The primary results in Tables 1 and 2 report mean &pm; std, but this standard deviation appears to be over different evaluation tasks, not over independent training runs. RL-based methods are sensitive to initialization and LLMs are nondeterministic. Without variability across multiple runs, we cannot know if the reported improvements are statistically significant.

**Reproducibility.** Many results rely on an internal, proprietary dataset. The authors claim that this dataset is more compelling due to its larger scale and complexity, but this means that the strongest results cannot be reproduced by others.

**Evaluation metrics.** The primary quality metrics are derived from an "`o4 mini` judge". The paper's own ablation reports only a moderate correlation with human labels, suggesting the judge is not a highly reliable proxy. Using simple ratios of "correct claims" is a statistically naïve approach for such a complex task. This evaluation method lacks the robustness of established psychometric or item-response theory models (e.g. Rasch models or paired comparisons) needed to assess generaiton quality.

**Presentation.** The paper is difficult to read and review due to several formatting and presentation issues. The main results tables are unreadably dense and poorly formatted, attempting to cram at least five different metrics and their associated variance scores into a single block. This makes the core findings very difficult for the reader to parse. Figure 1, the main architectural diagram, is also unreadable due to small font size and ineffective use of space. The paper consistently misuses in-text citations rather than parenthetical ones, presumably due to incorrect use of the `\cite` macro in LaTeX. For a general conference like ICLR, prior familiarity with multi-omics should not be assumed.

**Spurious precision.** The paper introduces mathematical notation that appears overly complex, such as the \\(R_\text{code}\\) function, which appears just to be an arbitrary concave function to map 3 successful actions to a reward of 1. The inclusion of specific metrics in the abstract similarly gives the impression of precision but serves to make the claims seem limited and dataset-specific.

**Questions:**

1. How many independent training runs were performed for each reported model in Tables 1, 2, 3, and A5? Please report the mean and standard error (or 95% CI) across these full runs.

2. Given the moderate \\(r=0.64\\) correlation with human labels, why should we trust the LLM-as-judge as the primary basis for your claims? Can you provide a more robust justification or, ideally, re-evaluate using a paired-comparison framework?

3. I understand the internal dataset is proprietary. However, can the evaluations on either the MLOmics or other open datasets not be expanded?

4. Can you provide a clear justification for the specific, complex mathematical form of the $R_{code}$ reward? What benefit does this specific log-based formula provide over a simpler clipped linear or exponential function?

5. The authors are strongly urged to re-format Tables 1 and 2 and re-render Figure 1 to be legible.

---

### Official Review · Reviewer_4Mod · 2025-11-01

**Soundness:** 2
**Presentation:** 2
**Contribution:** 2
**Rating:** 2
**Confidence:** 4

**Summary:**

The authors aim to develop agents that can summarize evidence from multi-table biomedical databases in response to natural language queries. They identify fluent yet overconfident responses as a key limitation of agents in this setting. To address this, the authors propose to integrate uncertainty signals into RL training and inference to encourage the model to have more accurate responses.

The agent environment for benchmarking consists of multiomics databases, tools to query the database schema and execute SQL and python, and to commit the final response. The authors propose training rewards with three components:
1. code correctness (increases up to 3 correct python / SQL executions)
2. "exploration judge reward": the count of "grounded, non-overlapping atomic facts" identified by o4-mini in the response (capped at 20)
3. "summary confidence reward": 1 / mean perplexity

They train agents with these rewards for 100 steps using two multiomics databases using curated tasks.

Model are evaluated based on:
1. number of claims in summary (LM as judge)
2. number of claims in summary (LM as judge, using 5 task specific workflows developed by experts)
3. ratio of useful claims (LM as judge)
4. whether uncertainty metrics (perplexity and semantic similarity across rollouts) are associated with higher ratio of incorrect claims

Key results: optimizing Qwen2.5-14B-instruct with proposed training setup yields improved evaluation metrics vs React / langchain agents with 4o-mini as well as ablated versions of their model without the perplexity reward. The authors observe improved survival time prediction using their optimized models.

**Strengths:**

The authors identify an important problem: using LLM's to effectively extract information from complex multi-omics databases. They use two interesting datasets, and define practical tasks. The survival task is particularly interesting due to its grounding in real data and clinical significance. Practical strategies to both improve training and filter responses are useful for establishing best practices for the community. The authors also make a good effort

**Weaknesses:**

1. I am not convinced by the experiments that adding uncertainty signals is the reason for improved performance. Notably, the LM-as-a-judge during training is not actually validating for correctness in the same sense as the validation, but instead querying o4-mini to count the number of atomic facts in the response. Thus, it seems the RL models without the perplexity reward do not have a clear correctness signal in the reward. The perplexity reward could also be interpreted as a an LM-as-a-judge correctness signal: it is rewarding outputs that the model predicts as more likely, which could implicitly reward responses aligned with the models pretraining knowledge. Under this interpretation, the ablation results are showing that an LM-as-a-judge correctness signal yields improvement in the task, which is different from the core argument of the paper.

2. 100 steps seems to be too short for experiments. Many of the models seem to be still improving in figure A5 at the end of the experiment, and it is not clear which methods are improving the ceiling of performance or just making the models converge more quickly

3. The evaluation criteria of counting the number and ratio of correct claims does not seem fully aligned with the use case. Should success not be based on a rubric of correctly answering the question(s) that are asked? Counting facts and looking at the ratio of correct components makes it difficult to interpret what is good performance: how many claims are actually required to address the query? Can performance be manipulated by whether some information is split into multiple claims?

4. The evaluation hinges heavily on the LM judge performance, which is difficult to interpret without prompts, example responses, and deeper analysis of the evaluation framework.

5. There are many ad-hoc choices in hyperparameters for the RL training (all models have code reward, specific coefficients used, ...). It is difficult to interpret the effect of these choices

5. The paper claims to be the first LLM agent framework to use uncertainty as a reward signal during training (which is implemented as using model perplexity of a response as a reward). For example, KL penalties take into account log-probs at the token level, and there are methods that use likelihood based rewards for calibrated uncertainty. I'm not familiar with other methods that directly use model perplexity as the reward which is interesting (but I may also be unfamiliar with methods that do this)

6. The introduction also mentions uncertainty as a data-quality signal as a key contribution, but this doesn't seem to be assessed in the experiments

**Questions:**

See weaknesses

---

### Note · Authors · 2025-11-12

I have read and agree with the venue's withdrawal policy on behalf of myself and my co-authors.